# Cortical softening elicits zygotic contractility during mouse preimplantation development

**Özge Özgüç**[1], **Ludmilla de Plater**[1], **Varun Kapoor**[1], **Anna Francesca Tortorelli**[1], **Andrew G. Clark**[2,3], **Jean-Léon Maître**[1]*

**1** Institut Curie, PSL Research University, Sorbonne Université, CNRS UMR3215, INSERM U934, Paris, France, **2** Institute of Cell Biology and Immunology, Stuttgart Research Center Systems Biology, University of Stuttgart, Stuttgart, Germany, **3** Center for Personalized Medicine, University of Tübingen, Tübingen, Germany

* jean-leon.maitre@curie.fr

**Citation:** Özgüç Ö, de Plater L, Kapoor V, Tortorelli AF, Clark AG, Maître J-L (2022) Cortical softening elicits zygotic contractility during mouse preimplantation development. PLoS Biol 20(3): e3001593. https://doi.org/10.1371/journal.pbio.3001593

**Data Availability Statement:** Data can be downloaded and used under a CC BY- NC-SA license at https://ressources.curie.fr/pecowaco/. The code used to analyze the oscillation

## Abstract

Actomyosin contractility is a major engine of preimplantation morphogenesis, which starts at the 8-cell stage during mouse embryonic development. Contractility becomes first visible with the appearance of periodic cortical waves of contraction (PeCoWaCo), which travel around blastomeres in an oscillatory fashion. How contractility of the mouse embryo becomes active remains unknown. We have taken advantage of PeCoWaCo to study the awakening of contractility during preimplantation development. We find that PeCoWaCo become detectable in most embryos only after the second cleavage and gradually increase their oscillation frequency with each successive cleavage. To test the influence of cell size reduction during cleavage divisions, we use cell fusion and fragmentation to manipulate cell size across a 20- to 60-μm range. We find that the stepwise reduction in cell size caused by cleavage divisions does not explain the presence of PeCoWaCo or their accelerating rhythm. Instead, we discover that blastomeres gradually decrease their surface tensions until the 8-cell stage and that artificially softening cells enhances PeCoWaCo prematurely. We further identify the programmed down-regulation of the formin Fmnl3 as a required event to soften the cortex and expose PeCoWaCo. Therefore, during cleavage stages, cortical softening, mediated by Fmnl3 down-regulation, awakens zygotic contractility before preimplantation morphogenesis.

## Introduction

During embryonic development, the shape of animal cells and tissues largely relies on the contractility of the actomyosin cortex [1–3]. The actomyosin cortex is a submicron thin layer of cross-linked actin filaments, which are put under tension by nonmuscle myosin II motors [4]. Tethered to the plasma membrane, the actomyosin cortex is a prime determinant of the stresses at the surface of animal cells [4,5]. Contractile stresses of the actomyosin cortex mediate crucial cellular processes such as the ingression of the cleavage furrow during cytokinesis [6–8], the advance of cells' back during migration [9,10], or the retraction of blebs [11,12]. At

frequencies from PIV and local curvature analyses can be found at https://github.com/MechaBlasto/PeCoWaCo.git. The Fiji plugin for local curvature analysis WizardofOz can be found under the MTrack repository.

**Funding:** JLM was funded by the Centre National de la Recherche Scientifique (CNRS); Institut National de la Santé et de la Recherche Médicale (Inserm); Fondation Schlumberger pour l'Education et la Recherche (FSER);EC | H2020 | H2020 Priority Excellent Science | H2020 European Research Council (ERC) ERC-2017-StG 757557; European Molecular Biology Organization - Young Investigator program; Université de Recherche Paris Sciences et Lettres (PSL) 17-CONV-0005; Université de Recherche Paris Sciences et Lettres (PSL) ANR-11-LABX-0044; Institut Curie. ÖÖ was funded by EC | H2020 | H2020 Priority Excellent Science | H2020 Marie Skłodowska-Curie Actions (MSCA) 666003; Fondation pour la Recherche Médicale (FRM) FDT202001010796; Institut Curie LP, VK, AFT were funded by Institut Curie. The funders had no role in study design, data collection and analysis, decision to publish, or preparation of the manuscript.

**Competing interests:** The authors have declared that no competing interests exist.

**Abbreviations:** hCG, human chorionic gonadotropin; IU, international unit; PeCoWaCo, periodic cortical waves of contraction; PIV, Particle Image Velocimetry; PMSG, pregnant mare serum gonadotropin; PMT, photomultiplier; TE, trophectoderm; WT, wild type; ZGA, zygotic genome activation; ZP, zona pellucida.

the tissue scale, spatiotemporal changes in actomyosin contractility drive apical constriction [13,14] or the remodeling of cell–cell contacts [15,16]. Although tissue remodeling takes place on timescales from tens of minutes to hours or days, the action of the actomyosin cortex is manifest on shorter timescales of tens of seconds [1,3,17]. In fact, actomyosin is often found to act via pulses of contraction during morphogenetic processes among different animal species from nematodes to human cells [13,14,18–25]. A pulse of actomyosin begins with the polymerization of actin filaments and the sliding of myosin minifilaments until maximal contraction of the local network within about 30 seconds [13,26,27]. Then, the actin cytoskeleton disassembles, and myosin is inactivated, which relaxes the local network for another 30 seconds [28–30]. These cycles of contractions and relaxations are governed by the turnover of the Rho GTPase and its effectors, which are well-characterized regulators of actomyosin contractility [19,29,31]. Indeed, the Rho pathway controls both the activity of myosin motors via their phosphorylation and the turnover of actin filaments via formins [5,19,26,29]. In instances where a sufficient number of pulses occur, pulses of contraction display a clear periodicity. The oscillation period of pulsed contractions ranges from 60 seconds to 200 seconds [14,18,19,22]. The period appears fairly defined for cells of a given tissue but can vary between tissues of the same species. What determines the oscillation period of contraction is poorly understood, although the Rho pathway may be expected to influence it [19,29,30]. Finally, periodic contractions can propagate into traveling waves. Such periodic cortical waves of contraction (PeCoWaCo) were observed in cell culture, starfish, and frog oocytes as well as in mouse preimplantation embryos [19,22,32,33]. In starfish and frog oocytes, mesmerizing Turing patterns of Rho activation with a period of 80 seconds and a wavelength of 20 μm appear in a cell cycle–dependent manner [19,34]. Interestingly, experimental deformation of starfish oocytes revealed that Rho activation wave front may be coupled to the local curvature of the cell surface [35], which was proposed to serve as a mechanism for cells to sense their shape [34]. In mouse embryos, PeCoWaCo with a period of 80 seconds were observed at the onset of blastocyst morphogenesis [22,36]. What controls the propagation velocity, amplitude, and period of these waves is unclear, and the potential role of such evolutionarily conserved phenomenon remains a mystery.

During mouse preimplantation development, PeCoWaCo become visible before compaction [22], the first morphogenetic movements leading to the formation of the blastocyst [3,37,38]. During the second morphogenetic movement, prominent PeCoWaCo are displayed in prospective inner cells before their internalization [36]. In contrast, cells remaining at the surface of the embryo display PeCoWaCo of lower amplitude due to the presence of a domain of apical material that inhibits the activity of myosin [36]. Then, during the formation of the blastocoel, high temporal resolution time-lapse hint at the presence of PeCoWaCo as microlumens coarsen into a single lumen [39]. Therefore, PeCoWaCo appear throughout the entire process of blastocyst formation [3]. However, little is known about what initiates and regulates PeCoWaCo. The analysis of maternal zygotic mutants suggests that PeCoWaCo in mouse blastomeres result primarily from the action of the nonmuscle myosin heavy chain IIA (encoded by *Myh9*) rather than IIB (encoded by *Myh10*) [40]. Dissociation of mouse blastomeres shows that PeCoWaCo are cell autonomous since they persist in single cells [22]. Interestingly, although removing cell–cell contacts free up a large surface for the contractile waves to propagate, the oscillation period seems robust to the manipulation [22]. Similarly, when cells form an apical domain taking up a large portion of the cell surface, the oscillation period does not seem to be different from cells in which the wave can propagate on the entire cell surface [36]. This raises the question of how robust PeCoWaCo are to geometrical parameters, especially in light of recent observations in starfish oocytes [34,35]. This question becomes particularly

relevant when considering that, during preimplantation development, cleavage divisions halve cell volume with each round of cytokinesis [41,42].

In this study, we investigate how the contractility of the cleavage stages emerges before initiating blastocyst morphogenesis. We take advantage of the slow development of the mouse embryo to study thousands of pulsed contractions and of the robustness of the mouse embryo to size manipulation to explore the geometrical regulation of PeCoWaCo. We discover that the initiation, maintenance, or oscillatory properties of PeCoWaCo do not depend on cell size. Instead, we discover a gradual softening of blastomeres with each successive cleavage, which exposes PeCoWaCo. This softening results in part from the reorganization of the actin cortex due to the down-regulation of the formin Fmnl3 during the first cleavage stages. Together, this study reveals how preimplantation contractility is robust to the geometrical changes of the cleavage stages during which the zygotic contractility awakens.

## Results

### PeCoWaCo during cleavage stages

PeCoWaCo have been observed at the 8-, 16-cell, and blastocyst stages. To know when PeCoWaCo first appear, we imaged embryos during the cleavage stages and performed Particle Image Velocimetry (PIV) and Fourier analyses (Fig 1A–1C, S1 Movie). We note that PeCoWaCo pause during mitosis (S2 Movie), similarly to pulsed contractions in fly neuroblasts [24], and we have therefore excluded from our analysis embryos during mitosis. This analysis reveals that PeCoWaCo are detectable in fewer than half of zygote and 2-cell stage embryos and become visible in most embryos from the 4-cell stage onward (Fig 1D, S1A–S1F Fig, S1 Table, S1 Data). Furthermore, PeCoWaCo only display large amplitude from the 4-cell stage onward (Fig 1B and 1C). Interestingly, the period of oscillations of the detected PeCoWaCo shows a gradual decrease from 150 seconds to 80 seconds between the zygote and 8-cell stages (Fig 1E, S1 Table, S1 Data). The acceleration of PeCoWaCo rhythm could simply result from the stepwise changes in cell size after cleavage divisions. Indeed, we reasoned that if the contractile waves travel at constant velocity, the period will scale with cell size and shape. This is further supported by the fact that PeCoWaCo are detected at the same rate and with the same oscillation period during the early or late halves of the 2-, 4-, and 8-cell stages (S1G and S1H Fig, S2 Table, S1 Data). Therefore, we set to investigate the relationship between cell size and periodic contractions.

### Cell size is not critical for the initiation or maintenance of PeCoWaCo

First, to test whether the initiation of PeCoWaCo in most 4-cell stage embryos depends on the transition from the 2- to 4-cell stage blastomere size, we prevented cytokinesis. Using transient exposure to Vx-680 to inhibit the activity of Aurora kinases triggering chromosome separation, we specifically blocked the 2- to 4-cell stage cytokinesis without compromising the next cleavage to the 8-cell stage (Fig 2A and 2B, S3 Movie). This causes embryos to reach the 4-cell stage with blastomeres the size of 2-cell stage blastomeres. At the 4-cell stage, we detect PeCoWaCo in most embryos whether they have 4- or 2-cell stage size blastomeres (Fig 2C, S3 Table, S1 Data). Furthermore, the period of oscillation is identical to 4-cell stage embryos in both control and drug-treated conditions (Fig 2D, S3 Table, S1 Data). Importantly, we do not measure any change in cell surface tension when treating embryos with Vx-680 indicating that the treatment does not seem to impact the overall mechanics of the actomyosin cortex (S2 Fig, S4 Table, S1 Data). This suggests that 4-cell stage blastomere size is not required to initiate PeCoWaCo in the majority of embryos.

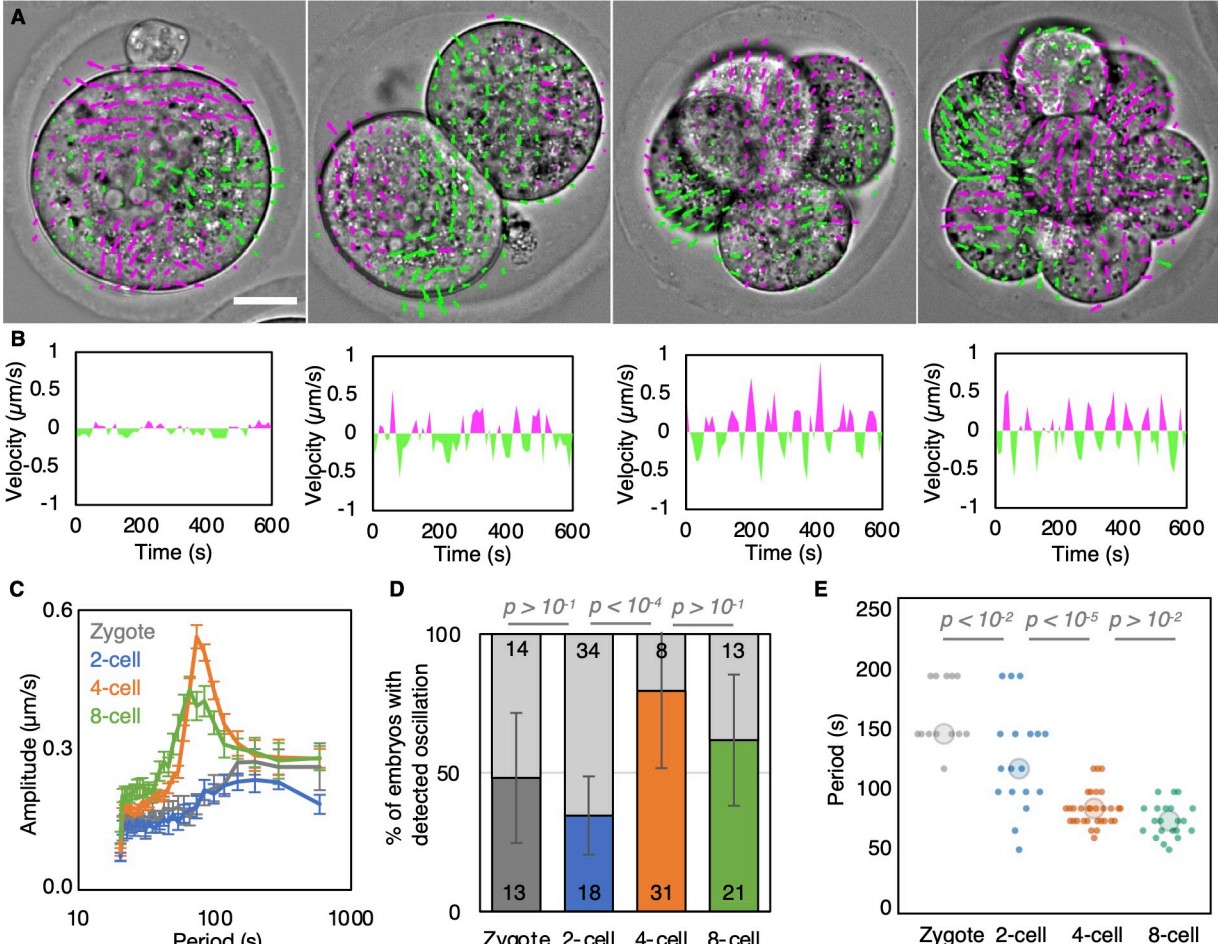

**Fig 1. Analysis of PeCoWaCo during cleavage stages. (A)** Representative images of a short-term time-lapse overlaid with a subset of velocity vectors from PIV analysis during cleavage stages (S1 Movie). Magenta for positive and green for negative Y directed movement. Scale bar, 20 μm. **(B)** Velocity over time for a representative velocity vector of each embryo shown in A. **(C)** Mean power spectrum resulting from Fourier transform of PIV analysis of zygote (gray, $n = 13$), 2-cell (blue, $n = 18$), 4-cell (orange, $n = 31$), and 8-cell (green, $n = 21$) stages embryos showing detectable oscillations. Data show as mean ± SEM (S1 Table). **(D)** Proportion of zygote (gray, $n = 27$), 2-cell (blue, $n = 52$), 4-cell (orange, $n = 39$), and 8-cell stage (green, $n = 34$) embryos showing detectable oscillations after Fourier transform of PIV analysis. Light gray shows nonoscillating embryos. Error bars show SEM. Chi-squared $p$-values comparing different stages are indicated (S1 Table, S1 Data). **(E)** Oscillation period of zygote (gray, $n = 13$), 2-cell (blue, $n = 18$), 4-cell (orange, $n = 31$), and 8-cell (green, $n = 21$) stages embryos. Larger circles show median values. Student $t$ test $p$-values are indicated (S1 Table, S1 Data). PeCoWaCo, periodic cortical waves of contraction; PIV, Particle Image Velocimetry.

Then, we tested whether PeCoWaCo could be triggered prematurely by artificially reducing 2-cell stage blastomeres to the size of a 4-cell stage blastomere. To reduce cell size, we treated dissociated 2-cell stage blastomeres with the actin cytoskeleton inhibitor Cytochalasin D before deforming them repeatedly into a narrow pipette (Fig 2E and 2F, S4 Movie). By adapting the number of aspirations of softened blastomeres, we could carefully fragment blastomeres while keeping their sister cell mechanically stressed but intact. Importantly, we measured identical surface tensions in intact and fragmented cells indicating that fragmentation does not seem to impact the overall mechanics of the actomyosin cortex (S2 Fig, S4 Table, S1 Data). While the fragmented cell was reduced to the size of a 4-cell stage blastomere, both fragmented and manipulated cells eventually succeeded in dividing to the 4-cell stage. After waiting 1 hour for cells to recover from this procedure, we examined for the presence of PeCoWaCo over the

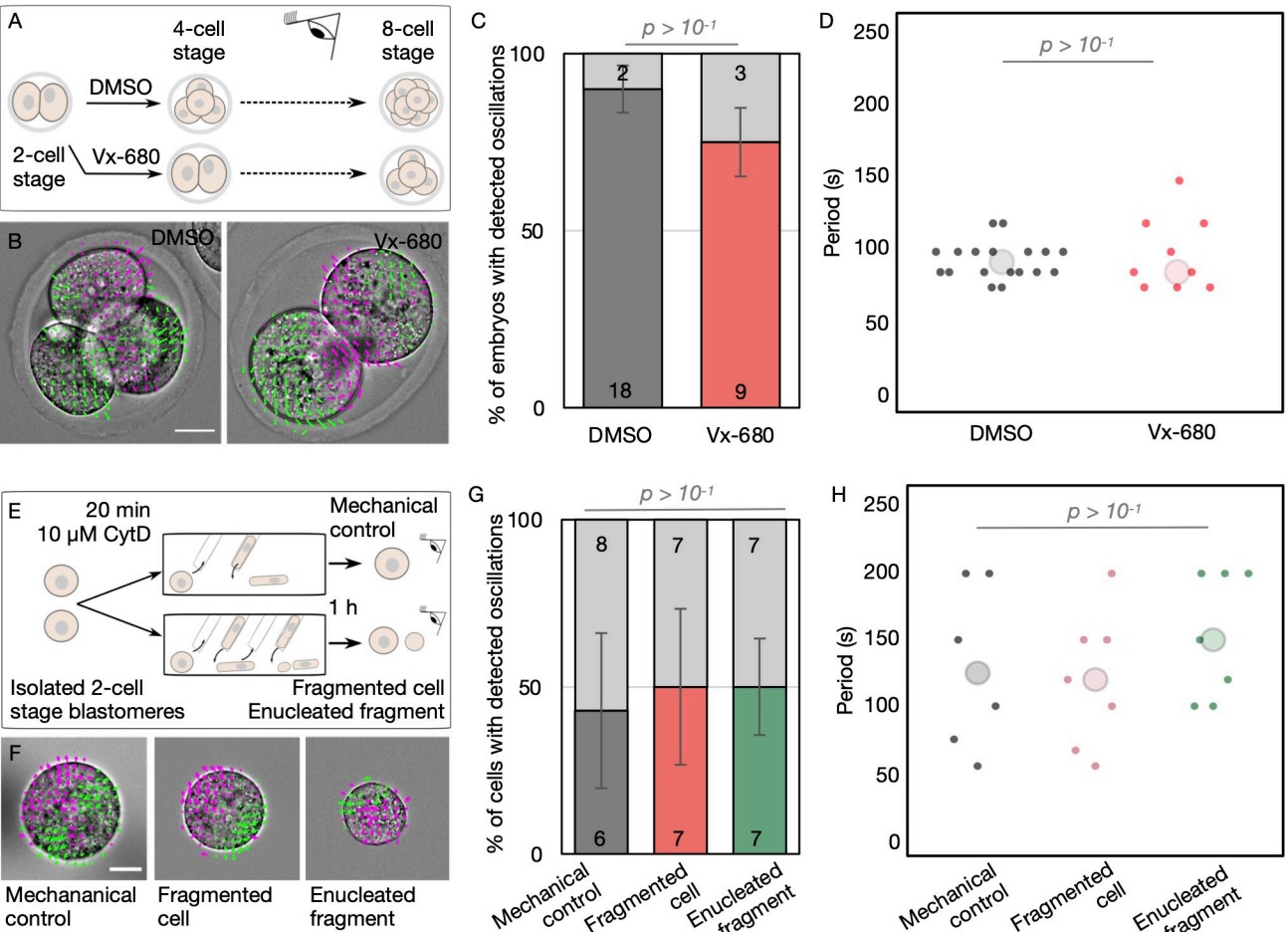

**Fig 2. Initiation of PeCoWaCo is independent of cell size. (A)** Schematic diagram of PeCoWaCo analysis after blocking the second cleavage division with 2.5 μM Vx680. **(B)** Representative images of DMSO and Vx-680 treated embryos overlaid with a subset of velocity vectors from PIV analysis (S3 Movie). Scale bar, 20 μm. **(C, D)** Proportion (C) of embryos showing detectable oscillations and their detected period (D, DMSO $n = 20$ and Vx-680 $n = 12$). Chi-squared (C) and Student $t$ test (D) $p$-values comparing 2 conditions are indicated (S3 Table, S1 Data). Error bars show SEM. Light gray shows nonoscillating embryos. Larger circles show median values. **(E)** Schematic diagram of PeCoWaCo analysis after fragmentation of 2-cell stage blastomeres. **(F)** Representative images of mechanical control, fragmented cell, and enucleated fragments overlaid with a subset of velocity vectors from PIV analysis (S4 Movie). Scale bar, 20 μm. **(G, H)** Proportion (G) of cells showing detectable oscillations and their detected period (H) in mechanical controls ($n = 14$), fragmented cells ($n = 14$), and enucleated fragments ($n = 14$). Error bars show SEM. Chi-squared (G) and Student $t$ test (H) $p$-values comparing 2 conditions are indicated (S3 Table, S1 Data). Light gray shows nonoscillating cells. PeCoWaCo, periodic cortical waves of contraction; PIV, Particle Image Velocimetry.

subsequent 10 hours. PeCoWaCo were detected in similar proportions in either control or fragmented cells (Fig 2G, S3 Movie, S3 Table, S1 Data). Also, the period of detected PeCoWaCo was unchanged (Fig 2H, S3 Table, S1 Data). This suggests that 4-cell stage blastomere size is not sufficient to trigger PeCoWaCo in the majority of embryos.

## Cell size does not influence the properties of PeCoWaCo

The transition from 2- to 4-cell stage blastomere size is neither required nor sufficient to initiate PeCoWaCo. Nevertheless, the decrease in period of PeCoWaCo remarkably scales with the stepwise decrease in blastomere size (Fig 1E). Given a constant propagation velocity, PeCoWaCo may reduce their period according to the reduced distance to travel around smaller cells. To test whether cell size determines PeCoWaCo oscillation period, we set out to

manipulate cell size over a broad range. For this, we used 16-cell stage blastomeres, which consistently display PeCoWaCo [36] and whose intermediate size permits broad size manipulation (Fig 3A–3D). By fusing varying numbers of 16-cell stage blastomeres, we built cells equivalent in size to 8-, 4-, and 2-cell stage blastomeres (Fig 3E–3G, S5 Movie, S5 Table, S1 Data) with undistinguishable surface tensions (S3A Fig, S6 Table, S1 Data). In addition, by fragmenting 16-cell stage blastomeres, we made smaller cells equivalent to 32-cell stage blastomeres (Fig 3H–3J, S6 Movie, S5 Table, S1 Data) [42]. Together, we could image 16-cell stage blastomeres with sizes ranging from 10 μm to 30 μm in radius (Fig 3K and 3L, S3B Fig). Finally, to identify how the period may scale with cell size by adjusting the velocity of the contractile wave, we segmented the outline of cells to compute the local curvature, which, unlike PIV analysis, allows us to track contractile waves and determine their velocity in addition to their period (Fig 3A–3D, see S4 Fig for comparisons between PIV and curvature analysis, S7 Movie) [22,36]. We find that fused and fragmented 16-cell stage blastomeres show the same period, regardless of their size (Fig 3F, 3I and 3K, S5 Table, S1 Data). This could be explained if the wave velocity would scale with cell size. However, we find that the wave velocity remains constant regardless of cell size (Fig 3E, 3J and 3L, S5 Table, S1 Data). Therefore, both the oscillation period and wave velocity are properties of PeCoWaCo that are robust to changes in cell size and associated curvature.

Fusion of cells causes blastomeres to contain multiple nuclei, while cell fragmentation creates enucleated fragments. Interestingly, enucleated fragments continued oscillating with the same period and showing identical propagation velocities as the nucleus-containing fragments (Fig 3I). These measurements indicate that PeCoWaCo are robust to the absence or presence of single or multiple nuclei and their associated functions.

Together, using fusion and fragmentation of cells, we find that PeCoWaCo oscillation properties are robust to a large range of size perturbations. Therefore, other mechanisms must be at play to regulate periodic contractions during preimplantation development.

## Cortical maturation during cleavage stages

Despite the apparent relationship between cell size and PeCoWaCo during preimplantation development, our experimental manipulations of cell size reveal that PeCoWaCo are not influenced by cell size. PeCoWaCo result from the activity of the actomyosin cortex, which could become stronger during cleavage stages and make PeCoWaCo more prominent as previously observed during the 16-cell stage [36]. Since actomyosin contractility generates a significant portion of the surface tension of animal cells, this would translate in a gradual increase in surface tension. To investigate this, we set to measure the surface tension of cells as a readout of contractility during cleavage stages using micropipette aspiration. Contrary to our expectations, we find that surface tension gradually decreases from the zygote to 8-cell stage (Fig 4A–4C, S7 Table, S1 Data) and noticeably mirrors the behavior of the period of PeCoWaCo during cleavage stages (Fig 1E). Therefore, PeCoWaCo unlikely result simply from increased contractility. Instead, the tension of blastomeres at the zygote and 2-cell stages may be too high for PeCoWaCo to become visible in most embryos. To reduce the tension of the cortex, we used low concentrations (100 nM) of the actin polymerization inhibitor Latrunculin A (Fig 4D and 4E, S7 Table, S1 Data) [32]. Softening the cortex of 2-cell stage embryos increased the proportions of embryos displaying PeCoWaCo (Fig 4F, S8 Movie, S7 Table, S1 Data). This suggests that PeCoWaCo become more visible thanks to the gradual softening of the cortex of blastomeres during cleavage stages. Moreover, low concentrations of Latrunculin A decreased the oscillation period of PeCoWaCo down to approximately 100 seconds, as compared to approximately 150 seconds for the DMSO control embryos (Fig 4G, S7 Table, S1 Data). This suggests

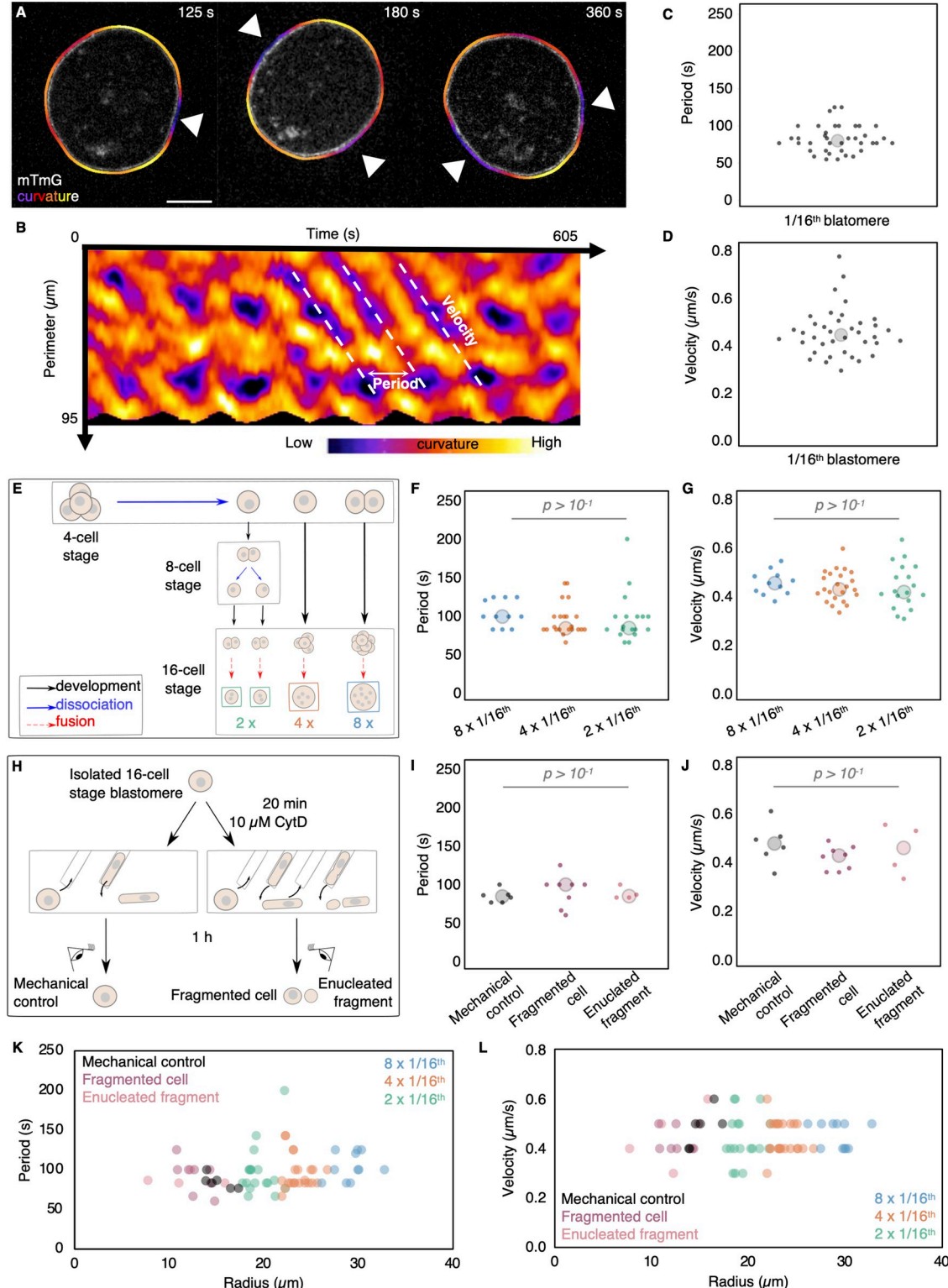

**Fig 3. Period and velocity of PeCoWaCo are stable across a broad range of cell sizes. (A, B)** Surface deformation tracking for period detection and velocity measurements. Isolated 16-cell stage blastomere originating from mTmG embryos with the local curvature fitted around it (A). Arrowheads indicate the PeCoWaCo. White scale bar, 10 μm. (B) Kymograph of curvature changes observed in the cell shown in (A). The period between consecutive waves and their velocity are indicated. Colored scale bar indicates curvature. **(C, D)** Period (C) and velocity (D) of PeCoWaCo in 38 isolated 16-cell stage blastomeres. Large circles show median. **(E)**

Schematic diagram of fusion of 16-cell stage blastomeres. **(F, G)** Oscillation period (F) and wave velocity (G) of fused blastomeres. 8 × 1/16th (blue, $n = 11$), 4 × 1/16th (orange, $n = 22$), and 2 × 1/16 (green, $n = 18$) fused blastomeres are shown. Large circles show median (S5 Movie, S5 Table, S1 Data). **(H)** Schematic diagram of fragmentation of 16-cell stage blastomeres. **(I, J)** Oscillation period (I) and wave velocity (J) of fragmented blastomeres. Control (black, $n = 6$), fragmented cell (magenta, $n = 8$), and enucleated fragment (pink, $n = 4$) are shown (S6 Movie, S5 Table, S1 Data). **(K, L)** Oscillation period (K) and wave velocity (L) for size-manipulated 16-cell stage blastomeres. Larger circles show median values. Student $t$ test $p$-values are indicated (S5 Table, S1 Data). CytD, Cytochalasin D; PeCoWaCo, periodic cortical waves of contraction.

that modifications of the polymerization rate of the actin cytoskeleton could be responsible for the increase in PeCoWaCo frequency observed during cleavage stages.

To investigate the changes responsible for blastomere softening during cleavage stages, we first looked into changes in cortical organization. Using super resolution microscopy on phalloidin-stained embryos, we measured the thickness of the actomyosin cortex, which has been reported to change with surface tension [43]. Using line scans orthogonal to the cell surface, we measured a width at half maximum of approximately 500 nm at the zygote stage (S5A–S5C Fig, S8 Table, S1 Data). This width increased during the 2-cell stage and fell back to its initial levels at the 4- and 8-cell stages (S5A–S5C Fig, S8 Table, S1 Data), suggesting cortical remodeling at the time of PeCoWaCo initiation.

To investigate the molecular changes responsible for blastomere softening during cleavage stages, we took advantage of available single-cell RNA sequencing and proteomic data [44]. We noted that several regulators of actin polymerization such as formins and actin related proteins (arps) decrease in their mRNA levels (S5D and S5E Fig). This is not necessarily the case for their protein levels (S5F and S5G Fig). This led us to investigate the formin Fmnl3, whose expression levels decay both at the mRNA and protein levels during cleavage stages (S5 Fig). Immunostaining of Fmnl3 finds it primarily at cell–cell contacts with levels seemingly decreasing during cleavage stages (S5H Fig). To test whether sustained expression of Fmnl3 would be sufficient to delay the appearance of PeCoWaCo, we injected embryos with mRNA encoding GFP-Fmnl3. Injection in a single blastomeres of 2-cell stage embryos resulted in visible overexpression of Fmnl3 at the 4-cell stage as observed using immunostaining (S6 Fig). Importantly, 4-cell stage blastomeres expressing GFP-Fmnl3 did not display PeCoWaCo like their uninjected counterparts (S6 Fig, S9 Movie). Consistently with mosaic sustained expression of GFP-Fmnl3, fewer embryos overexpressing GFP-Fmnl3 in all blastomeres showed PeCoWaCo as compared to those injected with GFP (Fig 4H–4J, S10 Movie, S7 Table, S1 Data). Interestingly, this effect is clearly present during the first half of the 4-cell stage embryos, but GFP-Fmnl3 expressing embryos recover almost to the levels of GFP expressing embryos by the second half of the 4-cell stage (Fig 4I and 4J, S7 Table, S1 Data). Over long-term development, embryos expressing GFP-Fmnl3 compacted normally and formed blastocysts (S11 Movie). Together, this indicates that overexpressing Fmnl3 has a specific but transient effect. In fly embryos, the effect of Fmnl overexpression was proposed to dampen oscillation by stiffening the cortex [26]. To test the effect of Fmnl3 overexpression on the mechanical properties of cleavage stage mouse embryos, we measured their surface tension. Indeed, we measured surface tensions twice higher for embryos expressing GFP-Fmnl3 than for those expressing GFP alone (Fig 4K, S7 Table, S1 Data). We conclude that Fmnl3 down-regulation during cleavage stages is required for the softening of the cortex, which elicits the appearance of PeCoWaCo.

Together, these experiments using the pulsatile nature of cell contractility reveal the unsuspected maturation of the cortex of blastomeres during the cleavage stages of mouse embryonic development.

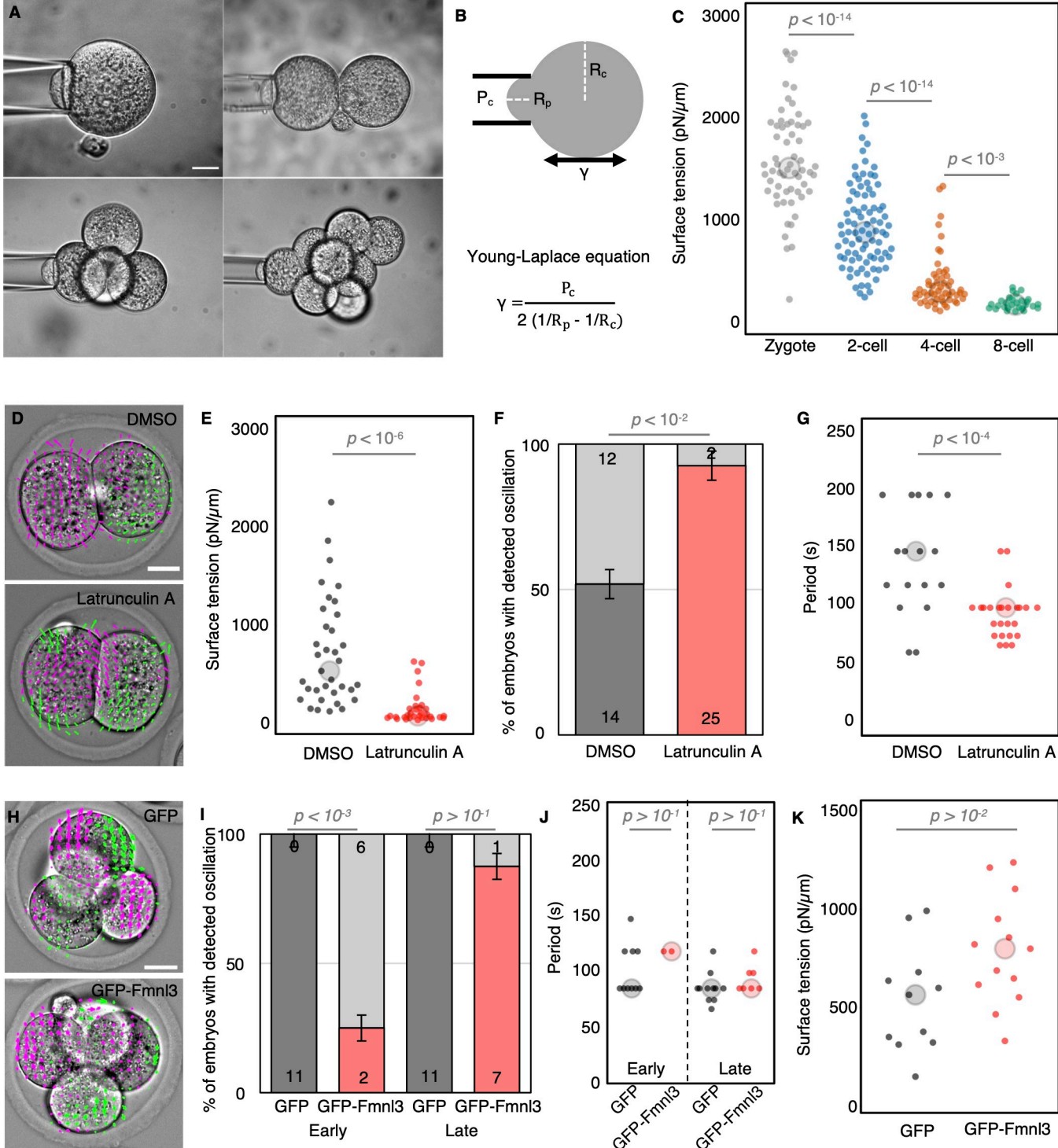

**Fig 4. Cortical softening elicits PeCoWaCo.** **(A)** Representative images of tension measurements at the zygote, 2-, 4-, and 8-cell stages. Scale bar, 20 μm. **(B)** Schematic diagram of the surface tension measurements. Using the Young–Laplace equation, the surface tension γ can be calculated from the critical pressure $P_c$ applied by a micropipette of radius $R_p$ onto a cell of radius of curvature $R_c$. **(C)** Surface tension of blastomeres throughout cleavage stages. Zygote (gray, $n = 60$), 2-cell (blue, $n = 86$), 4-cell (orange, $n = 55$), and early 8-cell (green, $n = 28$) stages are shown. Student $t$ test $p$-values are indicated (S7 Table, S1 Data). **(D)** Representative images of Control and 100 nM Latrunculin A treated embryos overlaid with a subset of velocity vectors from PIV analysis (S8 Movie). Scale bar, 20 μm. **(E)** Surface tension of embryos treated with DMSO ($n = 35$) or 100 nM Latrunculin A ($n = 32$). Student $t$ test $p$-value is indicated (S7 Table, S1 Data). Larger circles show median values. **(F, G)** Proportion (F) of embryos showing detectable oscillations and their detected period (G) of DMSO treated ($n = 27$) and 100 nM Latrunculin A treated ($n = 27$) 2-cell stage embryos. Error bars show SEM. Chi-squared (F) and Student $t$ test (G) $p$-values comparing 2

conditions are indicated (S7 Table, S1 Data). Light gray shows nonoscillating embryos. Larger circles show median values. **(H)** Representative images of embryos expressing GFP or GFP-Fmnl3 overlaid with a subset of velocity vectors from PIV analysis (S10 Movie). Scale bar, 20 μm. **(I, J)** Proportion (B) of embryos showing detectable oscillations and their detected period (C) in embryos expressing GFP (*n* = 11) or GFP-Fmnl3 (*n* = 8) at the early and late 4-cell stage. Error bars show SEM. Chi-squared (B) and Student *t* test (C) *p*-values comparing 2 conditions are indicated (S7 Table, S1 Data). Light gray shows nonoscillating embryos. **(K)** Surface tension of embryos expressing GFP (*n* = 11) or GFP-Fmnl3 (*n* = 13) measured during the early phase of the 4-cell stage. Student *t* test *p*-value is indicated (S7 Table, S1 Data). Larger circles show median values. PeCoWaCo, periodic cortical waves of contraction; PIV, Particle Image Velocimetry.

## Discussion

During cleavage stages, blastomeres halve their size with successive divisions. Besides the increased number of blastomeres, there is no change in the architecture of the mouse embryo until the 8-cell stage with compaction. We find that this impression of stillness is only true on a timescale of hours since, on the timescale of seconds, blastomeres display signs of actomyosin contractility. During the first 2 days after fertilization, contractility seems to mature by displaying more frequent and visible pulses. We further find that pulsed contractions do not rely on the successive reductions in cell size but rather on the gradual decrease in surface tension of the blastomeres. Therefore, during cleavage stages, cortical softening awakens zygotic contractility before preimplantation morphogenesis.

Previous studies on the cytoskeleton of the early mouse embryo revealed that both the microtubule and intermediate filament networks mature during cleavage stages. Keratin intermediate filaments appear at the onset of blastocyst morphogenesis [45] and become preferentially inherited by prospective trophectoderm (TE) cells [46]. The microtubule network is initially organized without centrioles around microtubule bridges connecting sister cells [47,48]. The spindle of early cleavages also organizes without centrioles similarly to during meiosis [47,49]. As centrioles form de novo, cells progressively transition from meiosis-like to mitosis-like divisions [47]. We find that the actomyosin cortex also matures during cleavage stages by decreasing its oscillation period (Fig 1E) and its surface tension (Fig 4C). Interestingly, this decrease in cortical tension seems to be in continuation with the maturation of the oocyte. Indeed, the surface tension of mouse oocytes decreases during their successive maturation stages [50]. The softening of the oocyte cortex is associated with architectural rearrangements that are important for the cortical movement of the meiotic spindle [51,52]. Therefore, similarly to the microtubule network, the zygotic actomyosin cortex awakens progressively from an egg-like state.

The maturation of zygotic contractility may be influenced by the activation of the zygotic genome occurring partly at the late zygote stage and mainly during the 2-cell stage [53,54]. Recent studies in frog and mouse propose that reducing cell size could accelerate zygotic genome activation (ZGA) [55,56]. We find that manipulating cell size is neither sufficient to trigger PeCoWaCo prematurely in most embryos nor required to initiate or maintain them in a timely fashion with the expected oscillation period of the corresponding cleavage stage (Figs 2 and 3). Instead, we find that the surface of blastomeres in the cleavage stages is initially too tense to allow for PeCoWaCo to be clearly displayed (Fig 4). We identify Fmnl3 down-regulation as an essential step in the reduction of blastomeres surface tension during cleavage stages (Fig 4). Taking place at the 2-cell stage, the contribution of the ZGA to zygotic contractility activation is unclear since Fmnl3 down-regulation begins after the zygote stage both at the mRNA and protein levels [44].

The effect of mechanical constraints on pulsed contractions is reminiscent of recent reports in fly embryos in which Fmnl-mediated densification of a persisting actin cytoskeleton dampens pulsed contractions [26]. In addition, the influence of mechanical constrains can also come from external structures. For example, in starfish oocytes, removing an elastic jelly

surrounding the egg softens them and renders contractile waves more pronounced [35]. Interestingly, the changes in surface curvature caused by cortical waves of contraction may influence the signaling and cytoskeletal machinery controlling the wave [35,57]. Such mechanochemical feedback has been proposed to regulate the period of contractions via the advection of regulators of actomyosin contractility [29]. As a result, the curvature of cells and tissues is suspected to regulate contractile waves [35,58]. Using cell fragmentation and fusion, we have manipulated the curvature of the surface over which PeCoWaCo travel (Fig 3). From radii ranging between 10 μm and 30 μm, we find no change in the period or traveling velocity of PeCoWaCo (Fig 3). This indicates that, in the mouse embryo, the actomyosin apparatus is robust to the changes in curvature taking place during preimplantation development. Therefore, the cleavage divisions per se are unlikely regulators of preimplantation contractility. The robustness of PeCoWaCo to changes of radii ranging from 10 μm to 30 μm is puzzling since neither the oscillation period nor the wave velocity seem affected (Fig 3K and 3L). One explanation would be that the number of waves present simultaneously changes with the size of the cells. Using our 2D approach, we could not systematically analyze this parameter. Nevertheless, we did note that some portions of the fused blastomeres did not display PeCoWaCo. These may be corresponding to apical domains, which do not show prominent PeCoWaCo [36]. Therefore, the relationship between the total area of the cell and the "available" or "excitable" area for PeCoWaCo may not be straightforward [19]. In the context of the embryo, in addition to the apical domain, cell–cell contacts also down-regulate actomyosin contractility and do not show prominent contractions [22]. As cell–cell contacts grow during compaction and apical domains expand [59,60], the available excitable cortical area for PeCoWaCo eventually vanishes [3].

Together, our study uncovers the maturation of the actomyosin cortex, which softens and speeds up the rhythm of contractions during the cleavage stages of the mouse embryo. Interestingly, zebrafish embryos also soften during their cleavage stages, enabling doming, the first morphogenetic movement in zebrafish [61]. It will be important to investigate whether cell and tissue softening during cleavage stages is conserved in other animals.

## Methods

### Embryo work

**Recovery and culture.**   All animal work is performed in the animal facility at the Institut Curie, with permission by the institutional veterinarian overseeing the operation (APAFIS #11054–2017082914226001). The animal facilities are operated according to international animal welfare rules.

Embryos are isolated from superovulated female mice mated with male mice. Superovulation of female mice is induced by intraperitoneal injection of 5 international units (IU) pregnant mare serum gonadotropin (PMSG, Ceva (CEVA Santé animale, Libourne, France), SYNCRO-PART), followed by intraperitoneal injection of 5 IU human chorionic gonadotropin (hCG, MSD Animal Health, Walton UK, Chorulon) 44 to 48 hours later. Embryos are recovered at E0.5 by dissecting in 37˚C FHM (LifeGlobal, Guildford, USA, ZEHP-050 or Millipore, Darmstadt, Germany, MR-122-D) from the oviduct the ampula, from which embryos are cleared with a brief (5 to 10 seconds) exposure to 37˚C hyaluronidase (Sigma, Saint Louis, USA, H4272).

Embryos are recovered at E1.5 or E2.5 by flushing oviducts from plugged females with 37˚C FHM using a modified syringe (Acufirm, 1400 LL 23).

Embryos are handled using an aspirator tube (Sigma, A5177-5EA) equipped with a glass pipette pulled from glass micropipettes (Blaubrand intraMark or Warner Instruments, Holliston, MA, USA).

Embryos are placed in KSOM (LifeGlobal, ZEKS-050 or Millipore, MR-107-D) or FHM supplemented with 0.1% BSA (Sigma, A3311) in 10 μL droplets covered in mineral oil (Sigma, M8410 or Acros Organics) unless stated otherwise. Embryos are cultured in an incubator with a humidified atmosphere supplemented with 5% $CO_2$ at 37°C.

To remove the zona pellucida (ZP), embryos are incubated for 45 to 60 seconds in pronase (Sigma, P8811).

For imaging, embryos are placed in 5- or 10-cm glass bottom dishes (MatTek, Ashland, MA, USA).

Only embryos surviving the experiments were analyzed. Survival is assessed by continuation of cell division as normal when embryos are placed in optimal culture conditions.

**Mouse lines.**  Mice are used from 5 weeks old on. (C57BL/6xC3H) F1 hybrid strain is used for wild-type (WT). To visualize plasma membranes, mTmG (Gt(ROSA)26Sor[tm4(ACTB-tdTomato,-EGFP)Luo]) is used [62].

**Isolation of blastomeres.**  ZP-free 2-cell or 4-cell stages embryos are aspirated multiple times (typically between 3 and 5 times) through a smoothened glass pipette (narrower than the embryo but broader than individual cells) until dissociation of cells.

For 8- and 16-cell stage embryos, they are placed into EDTA containing $Ca^{2+}$ free KSOM [63] for 8 to 10 minutes before dissociation. Cells are then washed with KSOM for 1 hour before experiment.

**Chemical reagents and treatments.**  Vx-680 (Tocris Bioscience, Bristol, UK, 5907) 50 mM DMSO stock was diluted to 2.5 μM in KSOM. To prevent mitosis, 2-cell stage embryos are cultured in 2.5 μM Vx-680 for 3 hours shortly prior to the second cleavage and then washed in KSOM. Subsequent divisions were observed in 12/20 of the cases.

Cytochalasin D (Sigma, C2618-200UL) 10 mM DMSO stock is diluted to 10 μM in KSOM. To fragment cells, isolated 2- or 16-cell stage blastomeres were treated with Cytochalasin D for 20 minutes before being gently aspirated into a smoothened glass pipette of diameter about 30 or 5 to 10 μm, respectively [59]. Moreover, 2 to 3 repeated aspirations are typically sufficient to clip cells into to 2 large fragments, one containing the nucleus and one without. Cells that did not fragment after 2 aspirations are used as control. For 2-cell stage fragmentation, nucleated fragments divisions were observed in 9/15 of the cases. Enucleated fragments started to deform extensively 6 hours after fragmentation, making it difficult to measure PeCoWaCo and surface tension.

GenomONE-CF FZ SeV-E cell fusion kit (Cosmo Bio, Tokyo, Japan, ISK-CF-001-EX) is used to fuse blastomeres [40]. HVJ envelope is resuspended following manufacturer's instructions and diluted in FHM for use. To fuse blastomeres of embryos at the 16-cell stage, embryos are incubated in 1:50 HVJ envelope for 15 minutes at 37°C followed by washes in KSOM.

Latrunculin A (Tocris Bioscience, ref 3973) 10 mM DMSO stock is diluted to 100 nM in KSOM. To soften cells, 2-cell stage embryos are imaged in medium containing Latrunculin A covered with mineral oil for 2 hours.

**Fmnl3 cDNA isolation, cloning, and in vitro mRNA synthesis.**  To isolate cDNA of *Fmnl3*, we performed total RNA extraction from a pool of 50 zygotes using the PicoPure RNA Isolation Kit (Thermo Fisher Scientific, Walthan, MA, USA, KIT0204). DNase treatment is performed during the extraction, using RNase-Free DNase Set (QIAGEN, Hilden, Germany, 79254). Subsequently, a cDNA library is synthesized with oligo(dT) (Thermo Fisher Scientific, 18418012) using the Super-Script III Reverse Transcriptase kit (Thermo Fisher Scientific, 18080044) on all the extracted RNA, according to manufacturer's instructions. As a final step, a fragment of 3,084 bp corresponding to *Fmnl3* isoform 202 (MGI:109569) is specifically isolated from the cDNA library, by PCR amplification with forward (fw) and reverse (rv) primers GCATGGACGAGCTGTACAAGGGCAACCTGGAGAGCACCGA and TAGTTCTAGACC GGATCCGGCTAACAGTTTGACTCGTCATG, respectively.

To generate the GFP-Fmnl3 plasmid construct for in vitro mRNA synthesis, the Gibson Assembly cloning method was used. Three linear DNA fragments, corresponding to pCS2 + backbone, *GFP* reporter gene, and *Formin like 3 (Fmnl3)* cDNAs, are initially generated by PCR amplification. During this step, overlapping ends are incorporated into each fragment.

Forward and reverse primers to obtain a 4,087-bp fragment of the pCS2+ backbone: TGACGAGTCAAACTGTTAGCCGGATCCGGTCTAGAACTATAGTGAGTCGT and AGTGAGTCGTATTACCGGATCCGGTCTATAGTGTCACCTAAATC.

Forward and reverse primers to obtain a 717-bp fragment encoding *GFP*: CGGTAATAC GACTCACTATAGGCCGGATCCGGATGGTGAGCAAGGGCGAGGA and TCGGTGCT CTCCAGGTTGCCCTTGTACAGCTCGTCCATGC.

Following DNA purification, the assembly of the final construct is achieved by incubating the 3 fragments in the Gibson Assembly Master Mix (NEB, Ipswich, MA, USA, E2611S), according to the manufacturer's instruction.

Following the linearization of the pCS2-GFP-Fmnl3 plasmid using Hind III, GFP-Fmnl3 mRNA is transcribed using the mMESSAGE mMACHINE SP6 Kit (Invitrogen, Waltham, MA, USA, AM1340) according to manufacturer's instructions and resuspended in Rnase-free water.

GFP mRNA is generated by in vitro transcription of a GFP linear DNA fragment of approximately 750 bp obtained by PCR amplification from the pCS2-GFP-Fmnl3 plasmid, with fw primer ATTTAGGTGACACTATAGAGCC and rv primer CTACTTGTACAGCTCGTC CAT.

**Microinjection.** Glass capillaries (Harvard Apparatus (Holliston, MA, USA) glass capillaries with 780-μm inner diameter) are pulled using a needle puller and micro forged to forge a holding pipette and an injection needle. The resulting injection needles are filled with mRNA solution diluted to 1 μg/μL in injection buffer (5 mM Tris-HCl pH = 7.4, 0.1 mM EDTA). The filled needle is positioned on a micromanipulator (Narishige MMO-4) and connected to a positive pressure pump (Eppendorf FemtoJet 4i). Embryos are placed in FHM drops covered with mineral oil under Leica (Wetzlar, Germany) TL Led microscope. Two-cell stage embryos were injected while holding with holding pipette connected to a Micropump CellTram Oil.

**Micropipette aspiration.** As described previously [22,64], a microforged micropipette coupled to a microfluidic pump (Fluigent, Le Kremlin-Bicêtre, France, MFCS EZ) is used to measure the surface tension of embryos. In brief, micropipettes of radii 8 to 16 μm are used to apply stepwise increasing pressures on the cell surface until reaching a deformation, which has the radius of the micropipette ($R_p$). At steady state, the surface tension $\gamma$ of the cell is calculated from the Young–Laplace's law applied between the cell and the micropipette: $\gamma = P_c / 2 (1/R_p - 1/R_c)$, where $P_c$ is the critical pressure used to deform the cell of radius of curvature $R_c$.

Eight-cell stage embryos are measured before compaction (all contact angles < 105˚), during which surface tension would increase [22].

Fragmented cells and their control cells are measured 10 to 15 hours after fragmentation. At that point, enucleated fragments are mostly irregular in shape and cannot be measured.

Measurements of individual blastomeres from the same embryo are averaged and plotted as such.

**Immunostaining.** Embryos are fixed in 2% PFA (Euromedex, Strasbourg, France, 2000-C) for 10 minutes at 37˚C, washed in PBS, and permeabilized in 0.01% Triton X-100 (Euromedex, T8787) in PBS (PBT) at room temperature before being placed in blocking solution (PBT with 3% BSA) at 4˚C for 2 to 4 hours. Primary antibodies (Table 1) are applied in blocking solution at 4˚C overnight. After washes in PBT at room temperature, embryos are incubated with secondary antibodies and phalloidin (Table 1) in blocking solution at room

**Table 1. Antibodies and dyes used for immunostaining.**

| Primary antibody | Dilution | Provider |
|---|---|---|
| Fmnl3 | 1:200 | [65] |
| **Secondary antibodies and dyes** | **Dilution** | **Provider** |
| Alexa Fluor 647 anti-guinea pig | 1:200 | Jackson ImmunoResearch, 706-605-148 |
| Alexa Flour 488 phalloidin | 1:500 | Thermo Fisher Scientific, A12379 |
| Alexa Flour 405 phalloidin | 1:200 | Thermo Fisher Scientific, A30104 |

temperature for 1 hour. Embryos are washed in PBT and imaged in PBS-BSA immediately after.

## Microscopy

For live imaging, embryos are placed in 5-cm glass bottom dishes (MatTek) under a CellDiscoverer 7 (Zeiss, Oberkochen, Germany) equipped with a 20×/0.95 objective and an ORCA-Flash 4.0 camera (C11440, Hamamatsu, Hamamatsu City, Japan) or a 506 axiovert (Zeiss) camera.

Using the experiment designer tool of ZEN (Zeiss), we set up nested time-lapses in which all embryos are imaged every 3 to 5 hours for approximately 10 minutes with an image taken every 5 seconds at 2 focal planes positioned 10 μm apart. Embryos are kept in a humidified atmosphere supplied with 5% $CO_2$ at 37˚C.

mTmG embryos are imaged at the 16-cell stage using an inverted Zeiss Observer Z1 microscope with a CSU-X1 spinning disc unit (Yokogawa, Tokyo, Japan). Excitation is achieved using a 561 nm laser through a 63×/1.2 C Apo Korr water immersion objective. Emission is collected through 595/50 band-pass filters onto an ORCA-Flash 4.0 camera (C11440, Hamamatsu). The microscope is equipped with an incubation chamber to keep the sample at 37˚C and supply the atmosphere with 5% $CO_2$.

Surface tension measurements are performed on a Leica DMI6000 B inverted microscope equipped with a 40×/0.8 DRY HC PL APO Ph2 (11506383) objective and Retina R3 camera and 0.7× lens in front of the camera. The microscope is equipped with an incubation chamber to keep the sample at 37˚C and supply the atmosphere with 5% $CO_2$.

Stained embryos are imaged on a Zeis LSM900 Inverted Laser Scanning Confocal Microscope with Airyscan detector. Excitation is achieved using a 488-nm laser line through a 63×/1.4 OIL DICII PL APO objective. Emission is collected through a 525/50 band-pass filter onto an airyscan photomultiplier (PMT) allowing to increase the resolution up to a factor 1.7.

## Data analysis

**Image analysis.** *Manual shape measurements*. Fiji [66] is used to measure cell, embryo, pipette sizes, and wave velocity. The circle tool is used to fit a circle onto cells, embryos, and pipettes. The line tool is used to fit lines onto curvature kymographs.

*PIV analysis*. To detect PeCoWaCo in phase contrast images of embryos, we use PIV analysis followed by a Fourier analysis.

As previously [22,40], PIVlab 2.02 running on MATLAB [67,68] is used to process approximately 10 minutes long time lapses with images taken every 5 seconds using 2 successive passes through interrogation windows of 20/10 μm resulting in approximately 180 vectors per embryo.

The x- and y-velocities of individual vectors from PIV analysis are used for Fourier analysis. A Fourier transform of the vector velocities over time is performed using MATLAB's fast Fourier transform function. The resulting Fourier transforms are squared to obtain individual

power spectra. Squared Fourier transforms in the x and y directions of all vectors are averaged for individual embryos resulting in mean power spectra of individual embryos.

Spectra of individual embryos are checked for the presence of a distinct amplitude peak to extract the oscillation period. The peak value between 50 seconds and 200 seconds was taken as the amplitude, as this oscillation period range is detectable by our imaging method. An embryo is considered as oscillating when the amplitude peaks 1.777 times above background (taken as the mean value of the power spectrum signal of a given embryo). This threshold value was determined using CutOffFinder [69] (S1 Fig) to minimize false positive and false negative according to visual verification of time-lapse movies. The number of oscillating zygote is likely overestimated, while the number of oscillating 8-cell stage is underestimated (S1 Fig).

Two-cell, 4-cell, and 8-cell stages are considered early during the first half of the corresponding stage and late during the second half.

Since PeCoWaCo halt during mitosis (S2 Movie), time lapses including dividing cells were excluded from the analysis.

*Local curvature analysis.* To measure PeCoWaCo period, amplitude, and velocity, we analyze the associated changes in surface curvature and perform Fourier analysis. Importantly, since we can only extract these parameters from oscillating blastomeres and embryos, data shown in Fig 3 and S4 Fig come from selected cells and embryos based on their visible oscillation.

To obtain the local curvature of isolated blastomeres and embryos, we developed an approach similar to that of [22,36,70]. First, a Gaussian blur is applied to images using Fiji [66]. Then, using ilastik [71], pixels are associated with cell surface or background. Segmentations of cells are then used in a custom made Fiji plug-in (called *WizardofOz*, found under the *Mtrack* repository) for computing the local curvature information using the start, center and end point of a 10-μm strip on the cell surface to fit a circle. The strip is then moved by 1 pixel along the segmented cell, and a new circle is fitted. This process is repeated till all the points of the cell are covered. The radius of curvature of the 10-μm strip boundaries are averaged. Kymograph of local curvature values around the perimeter over time is produced by plotting the perimeter of the strip over time.

Curvature kymographs obtained from local curvature tracking are then exported into a custom made Python script for 2D Fast Fourier Transform analysis.

Spectra of individual cells are checked for the presence of a distinct amplitude peak to extract the oscillation period. The peak value between 50 seconds and 200 seconds was taken as the amplitude, as this oscillation period range is detectable by our imaging method.

To measure the wave velocity, a line is manually fitted on the curvature kymograph using Fiji.

*Cortex thickness measurement.* Super resolution images obtained using airyscan microscopy are used to measure cortex thickness. The full width at half maximum of cortical intensity profiles were used to assess cortical thickness by using CortexThicknessAnalysis tool [43] available at https://github.com/PaluchLabUCL/CortexThicknessAnalysis.

**Statistics.** Data are plotted using Excel (Microsoft, Redmond, WA, USA) and R-based SuperPlotsOfData tool [72]. Mean, standard deviation, median, 1-tailed Student *t* test, and chi-squared *p*-values are calculated using Excel (Microsoft) or R (R Foundation for Statistical Computing). Statistical significance is considered when $p < 10^{-2}$.

The sample size was not predetermined and simply results from the repetition of experiments. No sample that survived the experiment, as assessed by the continuation of cell divisions, was excluded. No randomization method was used. The investigators were not blinded during experiments.

## Code availability

The code used to analyze the oscillation frequencies from PIV and local curvature analyses can be found at https://github.com/MechaBlasto/PeCoWaCo.git.

The Fiji plug-in for local curvature analysis *WizardofOz* can be found under the *MTrack* repository.

## Supporting information

**S1 Fig. (related to Fig 1). Analysis of PeCoWaCo during cleavage stages using PIV and Fourier transform. (A)** Peak value divided by mean signal of the respective power spectra after Fourier analysis of PIV data for embryos at the zygote, 2-, 4-, and 8-cell stages. Embryos are classified as showing a visible oscillation (+) or not (−), as assessed visually. **(B)** ROC curve resulting from CutOffFinder [62] showing that a threshold of 1.777 for the peak/mean amplitude (red cross) yields a maximized AUC of 0.97. **(C–F)** Classification of zygote (C), 2- (D), 4- (E), and 8-cell stages (F) using a threshold of 1.777 for the peak/mean amplitude. Red indicates a mismatch between the visual assessment and the use of the threshold to determine whether a given embryo oscillates or not at a given time point. **(G)** Proportion of early and late 2-cell (blue, $n = 33$ and 44), 4-cell (orange, $n = 38$ and 29), and 8-cell stage (green, $n = 17$ and 26) embryos showing detectable oscillations after Fourier transform of PIV analysis. Error bars show SEM. Chi-squared *p*-values comparing different stages are indicated (S2 Table, S1 Data). Light gray shows nonoscillating embryos. **(H)** Oscillation period of early and late 2-cell (blue, $n = 11$ and 9), 4-cell (orange, $n = 27$ and 17), and 8-cell (green, $n = 8$ and 15) stages embryos. Larger circles show median values. Student *t* test *p*-values are indicated (S2 Table, S1 Data). AUC, area under the curve; PeCoWaCo, periodic cortical waves of contraction; PIV, Particle Image Velocimetry; ROC, receiver operating characteristic.
(PDF)

**S2 Fig. (related to Fig 2). Surface tension of Vx-680 and fragmented embryos. (A)** Surface tension of embryos treated with DMSO ($n = 12$) or Vx-680 ($n = 13$). Student *t* test *p*-value is indicated (S4 Table). Larger circles show median values. **(B)** Surface tension of mechanical control ($n = 14$) or fragmented cells ($n = 14$). Student *t* test *p*-value is indicated (S4 Table). Larger circles show median values.
(PDF)

**S3 Fig. (related to Fig 3). Surface tension of fused blastomeres. (A)** Surface tension of cells resulting from the fusion of 8, 4, or 2 16-cell stage blastomeres ($n = 18, 20$ and 14 embryos, respectively). Student *t* test *p*-value is indicated (S6 Table, S1 Data). Larger circles show median values. **(B)** Radius of fused and fragmented 16-cell stage blastomeres. Larger circles show median values (S1 Data).
(PDF)

**S4 Fig. (related to Figs 1 and 3). Comparison of PIV and curvature analyses of PeCoWaCo. (A)** Representative images of a 4-cell stage embryo overlaid with a subset of velocity vectors from PIV analysis (top left) or a color coded local curvature analysis (top right). Velocity over time for a representative velocity vector (bottom left) or local curvature measurement (bottom right) for the bottom most blastomere. White scale bar, 10 μm. Colored scale bar indicates curvature. See S7 Movie. **(B)** Oscillation period of 4-cell stage embryos ($n = 6$) as determined by PIV or curvature tracking analyses. Pairwise Student *t* test *p*-value is indicated (S1 Data). PeCoWaCo, periodic cortical waves of contraction; PIV, Particle Image Velocimetry.
(PDF)

**S5 Fig. (related to Fig 4). Molecular and structural characterization of the actomyosin cortex during cleavage stages.** **(A)** Representative image of the actin cortex of a Phalloidin-stained 2-cell stage embryo acquired using super resolution microscopy. Scale bar, 5 μm. Overlaid white line shows the segmentation of the cortex, and orthogonal blue and yellow lines show line scans outside and inside the cell, respectively. Schematic diagram below shows example of where in the cell such image is taken. **(B)** Phalloidin intensity along lines orthogonal to the cell cortex of the cell shown in (A) (black, $n = 348$). Mean intensity profile shown in red. **(C)** Cortex thickness measured at the zygote, 2-, 4-, and 8-cell stage ($n = 15, 19, 13$, and 11, respectively). Student $t$ test $p$-values are indicated (S8 Table, S1 Data). Larger circles show median values. **(D–G)** mRNA (D and E) and protein (F and G) levels during cleavage stages for formins (D and F) and arps (E and G) adapted from [44]. **(H)** Immunostaining of FMNL3 (gray and cyan in merged image) on representative zygote, 2-, 4-, and 8-cell stage embryos. Phalloidin shown in magenta on the merged image. Scale bar, 20 μm.
(PDF)

**S6 Fig. (related to Fig 4). Molecular and structural characterization of the actomyosin cortex during cleavage stages.** **(A)** Schematic diagram of GFP-Fmnl3 mRNA injection in one blastomere of a 2-cell stage embryo. **(B)** Immunostaining of a 4-cell stage embryo injected with GFP-Fmnl3 at the 2-cell stage. GFP-Fmnl3 is shown on the left, and FMNL3 immunostaining is shown in gray in the middle and in cyan on the right together with Phalloidin in magenta. Scale bar, 20 μm. **(C)** Live imaging of a 4-cell stage embryo injected with GFP-Fmnl3 at the 2-cell stage. Left shows a merged image of phase contrast and GFP-Fmnl3. Right image shows the same embryo overlaid with a subset of velocity vectors from PIV analysis. Graphs on the right are velocity over time for a representative velocity vector of an uninjected (left) and GFP-Fmnl3 injected blastomere (right). Scale bar, 20 μm. See S9 Movie. **(D)** Schematic diagram of GFP or GFP-Fmnl3 mRNA injection in both blastomeres of a 2-cell stage embryo. **(E)** Representative images of the preimplantation development of GFP or GFP-Fmnl3 expressing embryos shown at the 4-, 8-, 16-cell, and blastocyst stages. Scale bar, 20 μm. S11 Movie. PIV, Particle Image Velocimetry.
(PDF)

**S1 Table. (related to Fig 1).** $p$-Values from chi-squared test for PeCoWaCo detection and from Student $t$ test for period comparisons. Red when above 0.05, green when below 0.01, and black in between. See S1 Data for individual quantitative observations. PeCoWaCo, periodic cortical waves of contraction.
(DOCX)

**S2 Table. (related to S1 Fig).** $p$-Values from chi-squared test for PeCoWaCo detection and from Student $t$ test for period comparisons. Red when above 0.05, green when below 0.01, and black in between. See S1 Data for individual quantitative observations. PeCoWaCo, periodic cortical waves of contraction.
(DOCX)

**S3 Table. (related to Fig 2).** $p$-Values from chi-squared test for PeCoWaCo detection and from Student $t$ test for period comparisons. Red when above 0.05, green when below 0.01, and black in between. See S1 Data for individual quantitative observations. PeCoWaCo, periodic cortical waves of contraction.
(DOCX)

**S4 Table. (related to S2 Fig).** $p$-Values from Student $t$ test. Red when above 0.05, green when below 0.01, and black in between. See S1 Data for individual quantitative observations.
(DOCX)

**S5 Table. (related to Fig 3).** *p*-Values from Student *t* test. Red when above 0.05, green when below 0.01, and black in between. See S1 Data for individual quantitative observations. (DOCX)

**S6 Table. (related to S3 Fig).** *p*-Values from Student *t* test. Red when above 0.05, green when below 0.01, and black in between. See S1 Data for individual quantitative observations. (DOCX)

**S7 Table. (related to Fig 4).** *p*-Values from chi-squared test for PeCoWaCo detection and from Student *t* test for period and surface tension comparisons. Red when above 0.05, green when below 0.01, and black in between. See S1 Data for individual quantitative observations. PeCoWaCo, periodic cortical waves of contraction. (DOCX)

**S8 Table. (related to S5 Fig).** *p*-Values from chi-squared test for PeCoWaCo detection and from Student *t* test for period and surface tension comparisons. Red when above 0.05, green when below 0.01, and black in between. See S1 Data for individual quantitative observations. PeCoWaCo, periodic cortical waves of contraction. (DOCX)

**S1 Data. Individual quantitative values collected to compute the data shown in figures.** Individual tabs are named according to the corresponding figure panels. (XLSX)

**S1 Movie. PIV analysis during cleavage stages.** Time-lapse imaging of zygote, 2-, 4-, and 8-cell stage embryos showing PeCoWaCo. Pictures are taken every 5 seconds, and PIV analysis is performed between 2 consecutive images. PIV vectors are overlaid on top of the images with vectors pointing upward in magenta and downward in green. Scale bar, 20 μm. PeCoWaCo, periodic cortical waves of contraction; PIV, Particle Image Velocimetry. (AVI)

**S2 Movie. PeCoWaCo are absent during mitosis.** Time-lapse imaging of 4- to 8-cell stage embryos before, during and after undergoing their third cleavage. Pictures are taken every 5 seconds. Scale bar, 20 μm. PeCoWaCo, periodic cortical waves of contraction. (AVI)

**S3 Movie. PIV analysis of 4-cell stage embryos treated with DMSO or Vx-680.** Time-lapse imaging of 4-cell stage embryos showing PeCoWaCo after treatment with DMSO or 2.5 μM Vx680 at the time of the second cleavage division. Pictures are taken every 5 seconds, and PIV analysis is performed between 2 consecutive images. PIV vectors are overlaid on top of the images with vectors pointing upward in magenta and downward in green. Scale bar, 20 μm. PeCoWaCo, periodic cortical waves of contraction; PIV, Particle Image Velocimetry. (AVI)

**S4 Movie. PIV analysis of fragmented 2-cell stage blastomeres.** Time-lapse imaging of mechanically manipulated and fragmented 2-cell stage blastomeres with and without nucleus. Pictures are taken every 5 seconds, and PIV analysis is performed between 2 consecutive images. PIV vectors are overlaid on top of the images with vectors pointing upward in magenta and downward in green. Scale bar, 20 μm. PIV, Particle Image Velocimetry. (AVI)

**S5 Movie. Surface deformation tracking of fused cells.** Montage of mTmG (top) and local curvature measurements (bottom) of fused 8×, 4×, 2× 1/16th blastomeres showing

PeCoWaCo. Scale bar, 20 μm. PeCoWaCo, periodic cortical waves of contraction.
(AVI)

**S6 Movie. Surface deformation tracking of fragmented 16-cell stage blastomeres.** Montage of mTmG (top) and local curvature measurements (bottom) of mechanically manipulated and fragmented 16-cell stage blastomeres with and without nucleus 1/16th blastomeres showing PeCoWaCo. Scale bar, 10 μm. PeCoWaCo, periodic cortical waves of contraction.
(AVI)

**S7 Movie. Comparison of PIV and curvature tracking analyses.** Time-lapse imaging of a 4-cell stage embryos. PIV analysis is performed between 2 consecutive images. PIV vectors are overlaid on top of the images with vectors pointing upward in magenta and downward in green. Curvature analysis is performed by locally fitting a circle to the cell surface. Pictures are taken every 5 seconds. Scale bar, 20 μm. PIV, Particle Image Velocimetry.
(AVI)

**S8 Movie. PIV analysis of 2-cell stage embryos treated with DMSO or Latrunculin A.** Time-lapse imaging of 2-cell stage embryos treated with DMSO or 100 nM Latrunculin A (LatA) showing PeCoWaCo. Pictures are taken every 5 seconds, and PIV analysis is performed between 2 consecutive images. PIV vectors are overlaid on top of the images with vectors pointing upward in magenta and downward in green. Scale bar, 20 μm. PeCoWaCo, periodic cortical waves of contraction; PIV, Particle Image Velocimetry.
(AVI)

**S9 Movie. PIV analysis of 4-cell stage embryo injected in one blastomere with GFP-Fmnl3 mRNA at the 2-cell stage.** Time-lapse imaging of 4-cell stage embryos expressing GFP-Fmnl3 in 2 blastomeres. First frame shows the merge of GFP-Fmnl3 (green) and phase contrast (gray) images. Pictures are taken every 5 seconds, and PIV analysis is performed between 2 consecutive images. PIV vectors are overlaid on top of the images with vectors pointing upward in magenta and downward in green. Scale bar, 20 μm. PIV, Particle Image Velocimetry.
(AVI)

**S10 Movie. PIV analysis of 4-cell stage embryo expressing GFP or GFP-Fmnl3.** Time-lapse imaging of 4-cell stage embryos expressing GFP or GFP-Fmnl3. Pictures are taken every 5 seconds, and PIV analysis is performed between 2 consecutive images. PIV vectors are overlaid on top of the images with vectors pointing upward in magenta and downward in green. Scale bar, 20 μm. PIV, Particle Image Velocimetry.
(AVI)

**S11 Movie. Preimplantation development of embryos expressing GFP or GFP-Fmnl3.** Time-lapse imaging of 4-cell stage embryos expressing GFP or GFP-Fmnl3. Pictures of GFP (green) and phase contrast (gray) are taken every 3 hours. Scale bar, 20 μm.
(AVI)

## Acknowledgments

We thank the imaging platform of the Genetics and Developmental Biology unit at the Institut Curie (PICT-IBiSA@BDD) for their outstanding support and the animal facility of the Institut Curie for their invaluable help. We thank the Matic Vignjevic, Plastino, Lennon-Duménil, and Bardin teams for sharing reagents. We thank Victoire Cachoux and Julie Firmin for help with image and data analysis. We are grateful to Ido Lavi, Sophie Louvet-Vallée, and members of

the Maître lab for discussions. We thank Julie Plastino and Markus Schliffka for critical reading of the manuscript.

## Author Contributions

**Conceptualization:** Özge Özgüç, Jean-Léon Maître.

**Data curation:** Özge Özgüç, Ludmilla de Plater, Varun Kapoor, Anna Francesca Tortorelli, Jean-Léon Maître.

**Formal analysis:** Özge Özgüç, Ludmilla de Plater, Varun Kapoor, Anna Francesca Tortorelli, Andrew G. Clark, Jean-Léon Maître.

**Funding acquisition:** Özge Özgüç, Jean-Léon Maître.

**Investigation:** Özge Özgüç, Jean-Léon Maître.

**Methodology:** Özge Özgüç, Ludmilla de Plater, Andrew G. Clark, Jean-Léon Maître.

**Project administration:** Jean-Léon Maître.

**Software:** Özge Özgüç, Varun Kapoor, Anna Francesca Tortorelli, Andrew G. Clark, Jean-Léon Maître.

**Supervision:** Özge Özgüç, Andrew G. Clark, Jean-Léon Maître.

**Validation:** Özge Özgüç, Jean-Léon Maître.

**Visualization:** Özge Özgüç, Jean-Léon Maître.

**Writing – original draft:** Özge Özgüç, Jean-Léon Maître.

**Writing – review & editing:** Özge Özgüç, Ludmilla de Plater, Varun Kapoor, Anna Francesca Tortorelli, Jean-Léon Maître.

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
