## [Editor Report · Decision Letter 0]

6 Oct 2021

Dear Dr Maitre, 

Thank you for submitting your manuscript entitled "Zygotic contractility awakening during mouse preimplantation development" for consideration as a Short Report by PLOS Biology.

Your manuscript, reviews from Review Commons, and revision plan have now been evaluated by the PLOS Biology editorial staff as well as by an Academic Editor with relevant expertise and I am writing to let you know that we would like to consider a revised version of your manuscript that addresses the reviewer comments from Review Commons.

Before we can invite you to submit a revised manuscript, we need you to complete your submission by providing the metadata that is required for full assessment. To this end, please login to Editorial Manager where you will find the paper in the 'Submissions Needing Revisions' folder on your homepage. Please click 'Revise Submission' from the Action Links and complete all additional questions in the submission questionnaire.

Once you have completed your submission, we will send you a formal "major revision" decision, which will include additional comments from the Academic Editor and a 3 month deadline for the revision.

Please re-submit your manuscript within two working days, i.e. by Oct 08 2021 11:59PM.

Given the disruptions resulting from the ongoing COVID-19 pandemic, please expect delays in the editorial process. We apologize in advance for any inconvenience caused and will do our best to minimize impact as far as possible.

Kind regards,

Lucas

Lucas Smith

Associate Editor

PLOS Biology

lsmith@plos.org

---

## [Editor Report · Decision Letter 1]

11 Oct 2021

Dear Dr Maitre,

Thank you for submitting your manuscript "Zygotic contractility awakening during mouse preimplantation development" for consideration as a Short Report at PLOS Biology. As mentioned in our last email, your manuscript, the reviews from Review Commons, and your revision plan have been evaluated by the PLOS Biology editors and an Academic Editor with relevant expertise.

In light of the reviews, which I have appended below, we will not be able to accept the current version of the manuscript. However, we would welcome re-submission of a much-revised version that takes into account the reviewers' comments as outlined in your submission. Having discussed your revision plan with the Academic Editor, we would like to emphasize the need to strengthen the evidence from the latrunculin experiments by a) providing measurements of changes in surface tension b) providing alternative methods to manipulate surface tension (including the Fmnl3, actin stability, and Rho and myosin inhibition experiments you propose). We also would emphasize the need to measure surface tension in size-manipulated embryos.

We cannot make any decision about publication until we have seen the revised manuscript and your response to the reviewers' comments. Your revised manuscript is also likely to be sent for further evaluation by the reviewers. 

We expect to receive your revised manuscript within 3 months. 

**IMPORTANT - SUBMITTING YOUR REVISION**

*Re-submission Checklist*

*Published Peer Review*

*PLOS Data Policy*

*Blot and Gel Data Policy*

Sincerely,

Lucas Smith

Associate Editor

PLOS Biology

lsmith@plos.org

REVIEWS:

Reviewer # 1

Evidence, reproducibility and clarity:

Periodic waves of contraction (PeCoWaCo) are stereotypical phenomena conserved in many oocytes and embryos that can lead to large-scale shape changes. However, their function is often unclear, and the parameters controlling their characteristics (de novo onset, amplitude, velocity...) still elusive in many model systems. In this manuscript, Ozguc and collaborators study the initiation of zygotic contractility during preimplantation development in the mouse. They show that PeCoWaCo become detectable in early embryos after the 2-cell stage, and progressively increase in frequency with each successive cleavage. Using elegant and challenging macro-manipulations of mouse embryos, they show that PeCoWaCo are not initiated by a reduction in cell size inherent in cell cleavage, nor their oscillation properties. Instead, PeCoWaCo are initiated and amplified by a gradual decrease in surface tension from the zygote to the 8-cell stage. The authors suggest that cortical softening awakens zygotic contractility.

The question addressed by the authors is an important one, in several fields (embryogenesis, cell biology, biophysics) and could therefore have wide impact. However, although the data presented are convincing, they fall short, controls are lacking to fully support the conclusions, and easy experiments could complement the existing limited results and take the paper to a higher level, making it a more complete study.

**Major comments:**

-Since the authors talk about "cortical maturation" during early embryogenesis and briefly about actin polymerization, they should describe the cortex itself during early embryogenesis (thickness, composition in terms of actin

nucleators, actin dynamics by FRAP...). This could also help to understand why embryos decrease their surface tension during early development.

-The authors use only one mean to reduce the tension of the cortex, using low doses of Latrunculin A. First, the authors should actually show that surface tension is reduced after this treatment by measuring it by micropipette aspiration, as they do in Figure 4 for control embryos, and show actin organization after such treatment (by immunofluorescence or live imaging). Second, could they use other means to reduce the tension of the cortex (for example play with actin polymerization by addressing actin nucleators/regulators at the cortex, or play with Myosin-II activity...)? Third, could they do the reverse, increase the tension of the cortex after the 2-cell stage to test if this inhibits/decreases PeCoWaCo (using Concanavallin A as in Kunda 2008, or embedding the embryos in loose collagen, or attracting

Myosin-II to the cortex...)? Finally, what is the long-term effect of disrupting the appearance/characteristics of PeCoWaCo on subsequent embryo development? This question could be addressed by tuning the intensity of PeCoWaCo using the experiments suggested above in the zygote or at the 2- cell stage, and monitoring embryo development up to the blastocyst stage to assess the outcome on embryo development.

-In most figures, with the exception of Figure 3, the authors detect and analyze PeCoWaCo using PIV, a technique that tracks movements in the cytoplasm. However, cytoplasmic movements are not always due to the contractility of the cortex and PeCoWaCo. They can be generated by cortex-independent cytoplasmic activity. Thus, it would have been preferable to follow the cell contour as in Figure 3 for all figures, for all stages (on dissociated embryos for

4 and 8-cell stages if too challenging due to overlapping of blastomeres in 2D). Another way to validate PIV as a means of detecting PeCoWaCo without having to repeat all the experiments could be to correlate PIV with oocyte contour in control embryos at all stages.

-In the treatments perturbing the cell size to test whether it is sufficient to initiate PeCoWaCo (Figure 2) or modify their periodicity (Figure 3), how is the surface tension between all conditions? Indeed, the authors nicely show that the cell size does not matter for PeCoWaCo, but that the cell surface does matter, so if PeCoWaCo are similar between cells of different sizes, it means that their surface tension should be in the same range as well. The authors should therefore measure it by micropipette aspiration.

- re the characteristics of the PeCoWaCo the same in interphase and mitosis? In Bischof 2017, surface contraction waves in starfish oocytes are regulated by a cdk1-cyclinB gradient in meiosis. Could such a mechanism play a role in what the authors observe in early mouse embryos? For example, is there a correlation between cell cycle length (and especially M-phase) and PeCoWaCo strength? If so (shorter cell cycles/M-phases in zygotes compared to 8-cell stage embryos), one could imagine that mouse embryos are progressively exposed to cdk1-cyclinB for longer periods of time during their development, which would contribute to the appearance and increase in PeCoWaCo.

**Minor comments:**

-Why does the proportion of embryos with oscillations decrease at the 8-cell stage compared to the 4-cell stage (Figure 1D)? It should not, since the surface tension is lower in 8-cell stage compared to the 4-cell stage (Figure 4B).

-Figure 2G shows that PeCoWaCo are not detected in either control or fragmented cells at the 2-cell stage but with a volume corresponding to a 4-cell stage, suggesting that 4-cell stage blastomere size is not sufficient to trigger PeCoWaCo. However, in Figure 1D, PeCoWaCo are detected in half of the control embryos at the 2-cell stage. Why is there a difference in the % of control embryos at the 2-cell stage with oscillations (0% versus 50%) between these 2 experiments? Could this be due to the CCD treatment allowing volume reduction (but also maybe reduction in surface tension) in Figure 2G?

-Since the authors fuse and dissociate embryos at the 16-cell stage, they should show the period and velocity of PeCoWaCo in unmanipulated 16-cell stage embryos prior to any treatment.

-In Figure 3, the percentage of embryos with detected oscillations should be added to the figure, as for Figure 1D or 2G.

-In Figure 3C-F, the period of the PeCoWaCo is about 100 s, higher than that of the 8-cell stage in Fig1E, which is about 75 s. Is this difference statistically significant? And if so, why would it be higher in Figure 3?

-Since PeCoWaCo become detectable after the 2nd cleavage, and thus after the ZGA, it could depend on a development program, allowing the surface tension to be reduced. The authors could have discussed this point further in the discussion

Finally, the figures are difficult to read and the legends lack precision:

-Some x-axis should be more informative with more than 3 values (Figures 1D,

2C, 2G, 4D, S1A).

- he value of the averages, or the exact percentages, should appear in full somewhere in the text, figures or legends (Figures 1D, 1E, 2C, 2D, 2G, 3C, 3D,

3F, 3G, 4B, 4D, 4E, S1A, S1B, S2).

-The exact p-value should be written in full somewhere in the text, figures or legends (concerns all figures).

-Sometimes the titles of the axes do not correspond to what the graph shows. For example, in graphs 1D, 2C, 2G, 4D, S1A, the x-axis shows all the embryos, not just the ones with oscillations.

-some figure legends lack information: What are the white arrows in Figure

3A? What are we looking at in Figure 3A, embryos in transmitted light or expressing a probe to follow the embryo contour? T and V in Figure 3A represent period and velocity, but it is not stated in the legend. The curvature bar in Figure 3A is strange as it is mostly white. The scale bar values are missing for Figures 2B, 2F, 4A and 4C.

-In Figure 3, the y-axes are not all consistent, sometimes the radius of the embryos is shown, sometimes fractions of 16-cell stage cells (which is not easy to read). 

SIGNIFICANCE: 

The question addressed by the authors is an important one, in several fields (embryogenesis, cell biology, biophysics) and could therefore have wide impact. However, although the data presented are convincing, they fall short, controls are lacking to fully support the conclusions, and easy experiments could complement the existing limited results and take the paper to a higher level, making it a more complete study.

Reviewer # 2

Evidence, reproducibility and clarity:

The manuscript by Özgüc and coworkers analyses the cortical contraction waves that are observed in early mouse embryos around the 8- to 16-cell stage. First, they provide a very careful, precise quantitative characterization of these contraction waves using PIV (particle image velocitometry) and by measuring the local cortex curvature. They find that contraction waves 'awaken' as development progresses, i.e. the oscillation frequency (and amplitude) of the waves is increasing during the cleavage divisions.

Then, they use a number of very elegant micromanipulations to dissect whether the decreasing cell size is the cause for the increased cortical activity and increased frequency of contraction waves. Somewhat surprisingly, but very clearly they demonstrate that cortical contractions are independent of cell size. These results are consistent across a wide range of manipulated cell (fragment) sizes, and various types of treatments.

Next, they measure the surface tension of cells using micropipette suction. These experiments reveal a clear correlation between surface tension and the period of contraction waves. To prove the causal relation between these processes, they artificially decrease the surface tension in 2-cell embryos by partially depolymerizing the cortex using Latrunculin A. This led to a strengthening of the contractions.

Together, the authors conclude that properties of cortical contraction waves are independent of cell size. Instead, they propose that the reason for the gradually increasing frequency of cortical contraction waves is the gradual softening of the cell cortex accompanying cleavage divisions.

**Major comments:**

1.The authors start introducing contraction waves by quantifying cytoplasmic flows using PIV. I would find it important to first show that deformations of the cortex are indeed causing the observed cytoplasmic flow patterns, which should be straightforward to analyze by using the tools shown on Fig. 3A. As there are other known, cortex independent mechanisms to generate cytoplasmic flows, these experiments would be important to exclude such alternative mechanisms.

2.I am very impressed by the various manipulations the authors use to demonstrate the independence of cell size and cortical contractions. However, I wonder why these manipulations have not been used in combination with surface tension measurements. Even more critically, I am missing more experiments that would directly demonstrate the causal relationship between cortical tension and contraction period. Unfortunately, the Latrunculin A results are less clear (as compared to other results shown in the paper), and the effect may indeed be more complex. For example, Bement et al (NCB, 2015) suggested a biochemical feedback mechanism based on F-actin, which in this experiment could not be distinguished from 'mechanical' effects of surface tension. Generally, I feel that additional perturbation experiments are needed in order to strengthen the conclusions. This should include upstream pathways known to regulate cortical contractility such as RhoA, for example.

**Minor comment:**

I would not mind if the authors used PeCoWaCo as an acronym through the present manuscript, but as they state, this is likely to be a very conserved feature of the cell cortex possibly of all animal cells. With this in mind, I doubt

that the community would adopt such a rather complicated acronym to describe this process.

SIGNIFICANCE:

Waves of cortical contractions are likely to be a widely conserved feature of the cortex of animal cells. The present manuscript makes important contributions to understand this process by (i) developing tools to quantify the process, (ii) show that contraction waves are independent of cell size, (iii) the period of contractions likely depend on the surface tension of the cell. These findings are broadly relevant for the cell biology community as a whole, however, as stated above, the findings needs to be substantially strengthened before publication, in my opinion.

---

## [Decision Letter · Decision Letter 2]

7 Feb 2022

Dear Dr Maitre,

Thank you for submitting your revised Short Reports entitled "Zygotic contractility awakening during mouse preimplantation development" for publication in PLOS Biology. I have now obtained advice from the original reviewers and have discussed their comments with the Academic Editor. 

Based on the reviews (attached below), we will probably accept this manuscript for publication, provided you satisfactorily address the remaining points raised by the reviewers. Please also make sure to address the following data and other policy-related requests.

Please also note that our Short Reports only have a maximum of 4 main figures, thus please redistribute the information currently shown in Figure 5 in the remaining figures or as a Supplementary Figure.

In addition, we would like to consider a suggestion to improve the title:

"Cortical softening mediated by Fmnl3 downregulation awakens zygotic contractility during mouse preimplantation development"

Please only use 'awakens' if it is an actual term used in the field. Otherwise, "elicits" would probably be more appropriate.

We expect to receive your revised manuscript within two weeks. 

*Published Peer Review History*

*Early Version*

Sincerely,

Ines

--

Ines Alvarez-Garcia, PhD,

Senior Editor,

ialvarez-garcia@plos.org,

PLOS Biology

Fig. 1B-E; Fig. 2C, D, G, F; Fig. 3C, D, F, G, I-L; Fig. 4C, E-G; Fig. 5B-D; Fig. S1A-H; Fig. S2A-B; Fig. S3A-B; Fig. S4A-B; Fig. S5B-G and Fig. S6C

Additionally, please provide all the accession numbers and make the files publicly available at this stage.

Reviewers' comments

Rev. 1:

As stated in my previous review for Review Commons, the manuscript has three major conclusions: (i) it provides a quantitative description of oscillatory cortical contraction waves in early mouse embryos; (ii) it shows using a series of perturbation and micromanipulation experiments that frequency and amplitude of these contraction waves is independent of cell size; (iii) it provides correlative evidence and a couple of perturbation experiments suggesting that progressive softening of the oocyte cortex is the cause of the observed 'awakening' of contractions as cleavage divisions proceed.

My major criticism was the lack of direct evidence for a causal relationship between cortical tension and contraction waves. This criticism was partially addressed by the authors in the revised manuscript by combining the perturbation experiment (low-dose latrunculin treatment) with measurement of the cortical tension, and by including an additional perturbation (overexpression of fmnl3). With these additions, I think the manuscript is very well suited for publication in PLoS Biology as a Short Report, as I consider the findings still somewhat preliminary, but certainly of high relevance, which is the exact purpose of the Short Reports, as I understand.

I only have a few minor comments to improve readability:

1. As I explained in my previous review, I do think it is very important to establish that PIV is a valid approach to report on cortical contractions. Therefore, I would suggest moving the panels now shown on FS4 into ideally Figure 1 or into one of the other main figures.

2. I would suggest to include the data on FS6 in the last main figure. I think this is a very a nice experiment!

3. I would suggest removing FS5 panel H, because without detailed validation these stainings are not easily interpretable.

Rev. 2:

The authors responded to all my comments by arguing or adding experiments, and the result is that the article is strengthened (both by adding controls, but also mechanistic like the part on Fmnl3).

One key question remains: what is the physiological role of these contractions ? In other word, what is the long-term effect of disrupting the appearance/characteristics of PeCoWaCo on the subsequent development of the embryo? This question could be addressed by tuning the intensity of PeCoWaCo in the zygote or at the 2-cell stage, and monitoring the development of the embryo up to the blastocyst stage to assess the outcome on embryo development. This question was only partly addressed by Fmnl3 overexpression, as the effect is transient: the

embryos show PeCoWaCo at the end of the 4-cell stage and develop to blastocyst normally. However, since this manuscript is considered a Short Report, this is not necessary and will pave the way for future research.

Minor:

- The scale bar values are still missing in the legends for Figures 2B, 2F, 4A, 4D 5A.

- The tables legends are wrong sometimes (Table 9 is related to figure S5 and not 5, Table 8 is related to figure 5 and not S5).

---

## [Editor Report · Decision Letter 3]

4 Mar 2022

Dear Jean-Leon,

On behalf of my colleagues and the Academic Editor, Sally Lowell, I am pleased to say that we can in principle accept your Short Report entitled "Cortical softening elicits zygotic contractility during mouse preimplantation development" for publication in PLOS Biology, provided you address any remaining formatting and reporting issues. These will be detailed in an email that will follow this letter and that you will usually receive within 2-3 business days, during which time no action is required from you. Please note that we will not be able to formally accept your manuscript and schedule it for publication until you have any requested changes.

PRESS

Sincerely, 

Ines

--

Ines Alvarez-Garcia, PhD 

Senior Editor 

PLOS Biology
